# Inferring reaction network structure from single-cell, multiplex data, using toric systems theory

**Shu Wang**[1,2], **Jia-Ren Lin**[1], **Eduardo D. Sontag**[1,3], **Peter K. Sorger**[1]*

**1** Laboratory of Systems Pharmacology, Department of Systems Biology, Harvard Medical School, Boston, Massachusetts, United States of America, **2** Harvard Graduate Program in Biophysics, Harvard University, Cambridge, Massachusetts, United States of America, **3** Departments of Electrical and Computer Engineering and of Bioengineering, Northeastern University, Boston, Massachusetts, United States of America

* peter_sorger@hms.harvard.edu

**Data Availability Statement:** All CyCIF data files are available from the Library of Integrated Network-based Cellular Signatures public database (http://lincs.hms.harvard.edu/db/datasets/20267/). The FACS data are available in the attached files of

## Abstract

The goal of many single-cell studies on eukaryotic cells is to gain insight into the biochemical reactions that control cell fate and state. In this paper we introduce the concept of Effective Stoichiometric Spaces (ESS) to guide the reconstruction of biochemical networks from multiplexed, fixed time-point, single-cell data. In contrast to methods based solely on statistical models of data, the ESS method leverages the power of the geometric theory of toric varieties to begin unraveling the structure of chemical reaction networks (CRN). This application of toric theory enables a data-driven mapping of covariance relationships in single-cell measurements into stoichiometric information, one in which each cell subpopulation has its associated ESS interpreted in terms of CRN theory. In the development of ESS we reframe certain aspects of the theory of CRN to better match data analysis. As an application of our approach we process cytometry- and image-based single-cell datasets and identify differences in cells treated with kinase inhibitors. Our approach is directly applicable to data acquired using readily accessible experimental methods such as Fluorescence Activated Cell Sorting (FACS) and multiplex immunofluorescence.

## Author summary

We introduce a new notion, which we call the effective stoichiometric space (ESS), that elucidates network structure from the covariances of single-cell multiplex data. The ESS approach differs from methods that are based on purely statistical models of data: it allows a completely new and data-driven translation of the theory of toric varieties in geometry and specifically their role in chemical reaction networks (CRN). In the process, we reframe certain aspects of the theory of CRN. As illustrations of our approach, we find stoichiometry in different single-cell datasets, and pinpoint dose-dependence of network perturbations in drug-treated cells.

the original Science article (DOI: 10.1126/science.1105809) in which they appear.

**Funding:** This work was supported by National Cancer Institute grant U54-CA225088 and Defense Advanced Research Projects Agency grant W911NF018-1-0124 (to PKS), Air Force Office of Scientific Research grant FA9550-14-1-0060 and NSF grant 1716623 and 1849588 (to EDS). SW was partially supported by National Institute of General Medical Sciences T32 GM008313. The funders had no role in study design, data collection and analysis, decision to publish, or preparation of the manuscript.

**Competing interests:** I have read the journal's policy and the authors of this manuscript have the following competing interests: PKS is a member of the SAB or Board of Directors of Merrimack Pharmaceutical, Glencoe Software, Applied Biomath, and RareCyte and has received research funding from Novartis and Merck. PKS declares that none of these relationships are directly or indirectly related to the content of this manuscript.

This is a *PLOS Computational Biology* Methods paper.

# Introduction

Single-cell, multiplex datasets have become prevalent [1, 2], and span a range of data types including transcript levels measured by sc-RNAseq [3], protein levels measured by flow cytometry [4], and cell morphology and protein localization measured by multiplex imaging [5–8]. An obvious advantage of such data is that it makes possible the detection and quantification of differences among cells in a population, including those arising from cyclic processes such as cell division, asynchronous differentiation programs [9, 10], and the effects of neighboring cells. A more subtle advantage of single-cell data is that they report on relationships among measured features, the phosphorylation states of receptors and nuclear localization of transcription factors for example. Because such features are subject to natural stochastic fluctuation across a population of cells [11], measuring the extent of correlation among otherwise independently fluctuating features makes it possible to infer the topologies of biological networks [12, 13].

A wide variety of tools have been developed for visualization of single-cell data, including t-SNE [14] and MAPPER [15], and for generating networks from such data using Bayesian Networks [16] and machine learning [17]. In many cases, the goal of such tools is to produce statistical models. In this paper we describe an alternative analytical framework founded on reaction theory. We make the assumption that proteins in a compartment react with each other in a manner that is well approximated by the continuum assumptions of Mass-Action Kinetics (MAK) [18], the foundation of familiar biochemical treatments of reactions such as Michaelis-Menten kinetics and Hill functions [19–21]. Compartments in this formalism can be different macromolecule assemblies or different locations in a cell. Cellular biochemistry is complicated, involving thousands of proteins and an unknown number of reaction compartments. Constructing dynamical systems of cellular processes based on MAK is computationally challenging, despite its theoretical appeal and analytical tractability. High-dimensional whole-cell dynamical models also suffer from a sparsity of data able to constrain such a model (although valuable insights have been obtained by this approach [22, 23]).

Unexpectedly, we have been able to sidestep some of the challenges posed by MAK in a cellular context by leveraging geometric aspects of dynamical systems and thereby obtaining analytical results from single-cell data. Chemical Reaction Network Theory (CRNT) is a branch of dynamical systems analysis that focuses primarily on topological features of a reaction network [24–26]. In this paper we frame results from CRNT in the context of single-cell, multiplex data. We demonstrate that unexpected insights into the topologies of reaction networks can be derived from such data based on familiar and simple MAK principles. Specifically, from multiplexed flow cytometry (FACS) and multiplexed immunofluorescence (CyCIF) data, we observe integer stoichiometry of reactions, and show that four anti-mitogenic drugs perturb a cell's reaction network in a mostly dose-independent manner.

To illustrate this approach, we briefly review some basic definitions. For $n$ chemicals $C_i$ involved in a reaction:

$$a_1 C_1 + a_2 C_2 + \cdots + a_n C_n \longrightarrow b_1 C_1 + b_2 C_2 + \cdots + b_n C_n$$

with reactants and products on the left and right respectively, and stoichiometric coefficients $\{a_i\}$ and $\{b_i\}$, we define a *reaction vector* $\vec{v} \in \mathbb{R}^n$ as:

$$\vec{v} = (b_1 - a_1, b_2 - a_2, \cdots, b_n - a_n).$$

Provided the reverse reaction exists, the *steady state* concentrations of the reactions obey the familiar equation:

$$\frac{\prod_{j=1}^{n} [C_j]^{b_j}}{\prod_{i=1}^{n} [C_i]^{a_i}} = K_{eq}$$

for some equilibrium constant $K_{eq}$. This equality can be rewritten in terms of $\vec{v}$, for the chemical concentrations $\vec{c} = ([C_1], [C_2], \cdots, [C_n])$:

$$\vec{v} \cdot \log(\vec{c}) = \log(K_{\text{eq}}), \tag{1}$$

where the logarithm of the vector is defined as the element-wise logarithm. (Logarithms are taken in any fixed basis, for example decimal). Observe that this is a linear equation on the reaction vectors, if one knows the (logarithms of) concentrations.

Given a network $G$ composed of such reactions, the overall dynamics are described by a system of differential equations, in which the rate of change of any chemical species' concentration is given by the sum of reaction rates in which it is a product, minus the sum of reaction rates in which it is a reactant [18]. As a simple example, consider a system of reactions:

$$X + Y \underset{k_{-1}}{\overset{k_1}{\rightleftarrows}} Z$$

$$\frac{d[X]}{dt} = \frac{d[Y]}{dt} = -\frac{d[Z]}{dt} = k_{-1}[Z] - k_1[X][Y].$$

We will focus on two objects associated with such systems: 1) the *steady state set* $\mathcal{E}$, defined as the set of concentrations for which all the time derivatives vanish, and 2) the *stoichiometric subspace S*, defined as the linear span of all the reaction vectors [27], which is a simple calculation for any given network. In the example described above, the steady state set is a nonlinear surface, shown in Fig 1a for $k_1 = k_{-1} = 1$, and its one-dimensional stoichiometric subspace is represented by a yellow line. The surface characterizes the network well, since any initial concentration ($[X], [Y], [Z]$) (denoted by red dots) approaches the steady state set. In our studies, $\mathcal{E}$ will be determined from experimental data, and we will be interested in reconstructing $S$, parts of $S$, or other subspaces of a similar stoichiometric nature. Which of these can be reconstructed depends on the class of network that is assumed to generate the data.

More specifically, among MAK dynamical systems, the subset known as "complex-balanced" reaction networks (which includes the familiar case of "detailed-balanced networks" [28]), has steady state sets that are easily expressed in terms of the stoichiometric subspaces [24]. Complex balancing means that each "complex" (a node of the reaction network, such as "$X + Y$" and "$Z$" in our example) is balanced with respect to inflow and outflow, analogous to a Kirchhoff current law (in-flux = out-flux, at each node). It is a nontrivial fact that, for every $\vec{v}_i \in S$, the steady state set $\mathcal{E}$ is precisely the set of all those vectors $\vec{c}$ that satisfy the following equalities:

$$\vec{v}_i \cdot \log(\vec{c}) = \log(K_i), \qquad i = 1, 2, \ldots. \tag{2}$$

This is analogous to the case of a single reaction in Eq 1, except that $K_i$ is not the equilibrium constant of the isolated reaction, but is instead a constant that accounts for kinetic constants from the entire network. The satisfaction of these equalities implies that, in log-concentration space, the transformed steady state set, $\log(E) \equiv V$, is an affine (linear with shift) subspace whose orthogonal complement coincides with $S$. Our earlier example was complex-balanced, so after taking the logarithm, its steady state surface becomes a plane in Fig 1b, whose orthogonal complement, in orange, is parallel to the yellow line, shown in Fig 1a. As

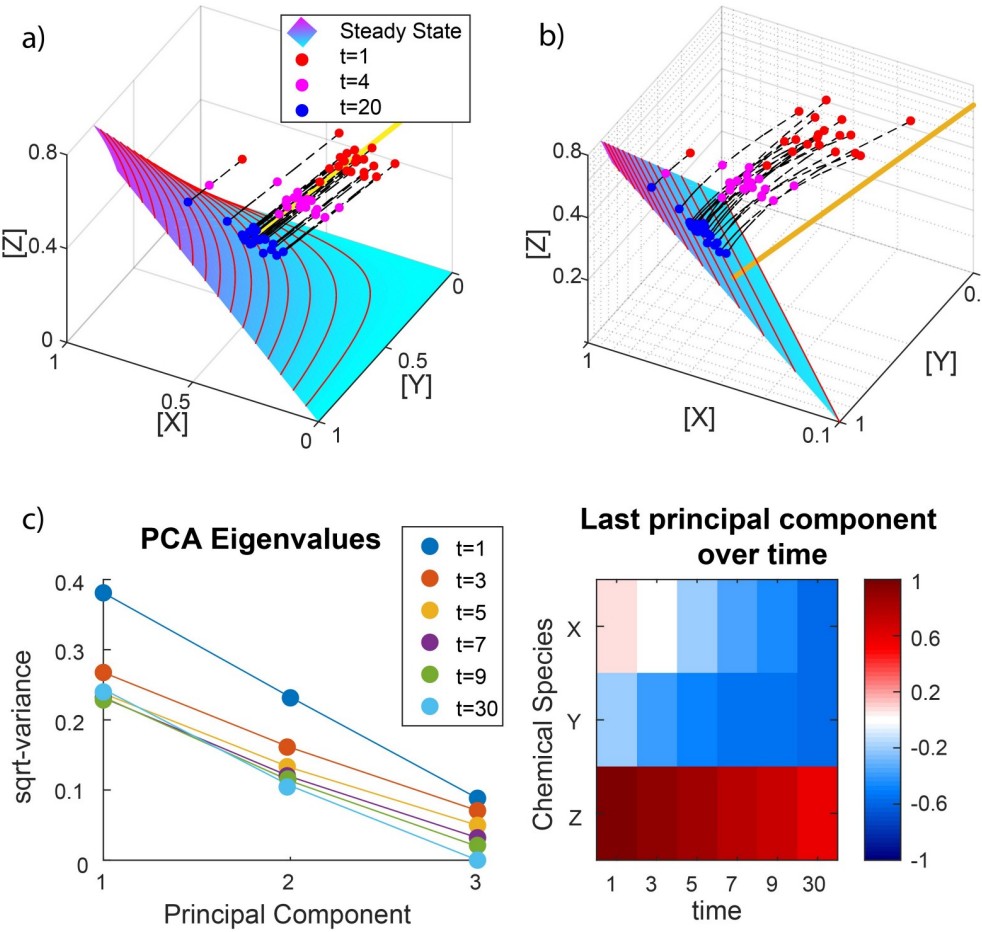

**Fig 1. MAK dynamical systems and covariance in single-cell-data.** (a) Several simulated trajectories of the reaction network $X + Y \rightleftharpoons Z$ are shown, for $k_1 = k_{-1} = 1$. The steady state set is shown in cyan/magenta, along with some of its level sets for fixed values of $[Z]$. A particular parallel translate of the stoichiometric subspace (coset) is shown as a yellow line. (b) Steady state set in logarithmic coordinates. The orthogonal complement of this subspace (orange) is parallel to the stoichiometric subspace. (c) In log-concentration space, the covariance matrix of the chemical trajectories will have a decreasing eigenvalue for $t \to \infty$, as evaluated by PCA, and the corresponding eigenvector will converge to the orthogonal complement, which is parallel to the stoichiometric subspace spanned by (-1,-1,1).

another example of a complex-balanced reaction network, consider a network with reversible and irreversible reactions:

$$
\begin{array}{c}
A \\
k_1 \downarrow \qquad \nearrow k_3 \\
B \underset{k_{-2}}{\overset{k_2}{\rightleftharpoons}} C
\end{array}
\tag{3}
$$

that has a steady state set satisfying (see S1 Appendix):

$$
\frac{[A]}{[B]} = \frac{k_1(k_{-2} + k_3)}{k_2 k_3}, \qquad \frac{[B]}{[C]} = \frac{k_2}{k_{-2} + k_3}, \qquad \frac{[C]}{[A]} = \frac{k_3}{k_1},
$$

which are log-linear relations in the form of Eq 2. Although the above two examples are always

complex-balanced, any network of any size can be complex-balanced if the kinetic constants $k_i$ are additionally constrained [25].

More generally, the subset of reaction networks that obey log-linearity are called *toric* in the algebraic-geometric CRNT literature [25, 29]. This log-linearity greatly simplifies the analysis of a nonlinear problem, which is the key appeal of our making a MAK assumption. The current work is concerned with such *toric* systems, of which complex-balanced reaction networks are the best-known example.

# Results

## Overview of the approach

We represent a single cell by a vector that includes as components the concentrations $\vec{c}_i(t)$ of relevant chemical species. We assume that all cells in the population being studied are governed by a common, complex-balanced, MAK reaction network $G$ with reaction constants $\{k_G\}$. The localization of a reactant into different cellular compartments (e.g. nucleus and cytoplasm) or different macro-molecular assemblies is managed using the conventional compartmentalized formalism and simply adds elements to $\vec{c}_i(t)$. As we will see, the fact that $G$ is incredibly complicated does not limit our theoretical analysis.

We reframe the equations described in the introduction in terms of the distribution of chemical trajectories from a population of cells, $p_G(\log(\vec{c}), t)$, making it possible to approximate the stoichiometric subspace $S$ of $G$ from a fixed-time sample distribution $p_G(\log(\vec{c}), t_{\text{fixed}})$, where $t_{\text{fixed}}$ is large in an appropriate sense. Typically, it is only possible to observe a subset of the species in a cellular reaction network. We find that when only a subset of the chemical species are observed, $\vec{c}_{\text{obs}}$, the covariance of $p_G(\log(\vec{c}_{\text{obs}}), t_{\text{fixed}})$ still makes it possible to determine a subset of $S$.

Exploring non-complex-balanced networks by simulation and examples, we find that our analysis method still recovers subspaces tied to reaction network topology, analogous to how $S$ is tied to reaction vectors. We call these data-derived subspaces Effective Stoichiometric Spaces (ESS), with the precise definition given below. The key extension to the complex-balanced case is that certain reaction networks that are non-complex-balanced can still have steady state sets contained in toric manifolds (either exactly or approximately), whose orthogonal complements in log-concentration space are related to reaction network topology (i.e. independent of kinetic parameters) [29–31].

With this theoretical background, we show that single-cell, multiplex data (sc-data) that can feasibly be obtained from mammalian cells using multiplexed flow cytometry (FACS) or multiplexed immunofluorescence (using CyCIF [5] and other similar methods) can be effectively analyzed within our framework, just using MAK assumptions. In particular, we find that (i) Principal Components Analysis (PCA) of single-cell data generates principal components (PCs) that lie on near-integer subspaces, which our framework interprets as the stoichiometric constants in the underlying reaction, and (ii) for cells exposed to different small molecule inhibitors of regulatory proteins (primarily protein kinase inhibitors), the covariance structure is conserved over a range of concentrations for any inhibitor, which our framework explains as the conservation of reaction network topology.

## Single-cell covariance from complex-balanced reaction networks

Suppose that a population of $N$ chemical trajectories $\vec{c}_i(t)$ is governed by a complex-balanced, MAK reaction network $G$, with stoichiometric subspace $S$ and steady state set $\mathcal{E}$. As $t \to \infty$,

$p_G(\log(\vec{c}), t)$ approaches a distribution supported on $V \equiv \log(\mathcal{E})$, whose sample covariance matrix $\Sigma$ is a singular matrix with singular eigenspace equal to $S$ (see Methods).

Applied to the example in Fig 1, the trajectories of $X$, $Y$, and $Z$ concentrations at any time $t$ constitute a dataset whose sample covariance matrix has one eigenvalue approaching zero as $t \to \infty$ (Fig 1c). This eigenvalue's corresponding eigenvector approaches $(-1, -1, 1)$, whose span is the stoichiometric subspace represented earlier by the orange line in Fig 1b.

In general, if we identify each cell in a population with a vector for the concentrations of all its relevant biochemical species $\vec{c}_i(t)$, the hypotheses above allow us to extract the stoichiometric subspace $S$ of the underlying reaction network by eigendecomposition of the sc-data covariance. This computation is commonly performed by PCA [32]. Whereas most applications of PCA focus on PCs that explain the greatest variance (e.g. PC1-3), we are interested in the singular eigenspace $S$, which is spanned approximately by the principal components that explain the least variance. To identify $S$ with real data we look for a gap in the eigenvalue spectrum: the eigenvalues converging to 0 will be small and similar in magnitude, forming a cluster, while the remaining, larger eigenvalues will appear separate from that cluster. When such eigenvalues are arranged in ascending order, a gap appears right after the last eigenvalue of the small cluster. Such a gap is unexpected under the null hypothesis that the data is drawn from a random, multivariate normal distribution with equal variance in all directions [33]. Finding a gap in real data is nontrivial, and we discuss this subtlety in later sections when we analyze FACS data.

## Timescale separation

For finite but sufficiently long times $t$, information about timescales can be found in sc-data. The eigenvalue spectrum of $\Sigma$, under the hypotheses described above, has at least one "gap"—a region of nonuniform spacing between neighboring eigenvalues—which separates the eigenvalues into "small" and "large" values. The small eigenspace approaches $S$ as $t$ increases. Additional gaps may indicate embedded subspaces $S_i \subset S$, spanned by the reactions that occur on faster timescales, so that we have $S_1 \subset S_2 \subset \ldots \subset S$ corresponding to different cutoffs for "fast" and "slow" (see Methods).

Following our earlier example of $X + Y \rightleftharpoons Z$ in Fig 1, we add a reaction $X \rightleftharpoons Y$ with forward and reverse reaction constants $k_f = 0.2$, $k_r = 0.1$, much slower than the original reaction. The trajectories now converge first to the earlier surface, since it is the steady state of the fast reaction. With enough time, those trajectories eventually converge to the steady state of both reactions (see Fig 2a), which is now a curve embedded in the surface. This separation of timescales is studied formally using *singular perturbation theory* [34] for dynamical systems, in which the first surface is the *slow manifold*, because trajectories converge quickly to its neighborhood, before undergoing slow dynamics constrained to that neighborhood.

For detailed-balanced reaction networks, slow manifolds are approximately the steady state sets of fast networks defined by ignoring slow reactions [35], and one might expect this to be true more generally. If so, then just as a single gap would appear when trajectories converge to the full network's steady state manifold, a gap appears as trajectories converge to the fast network's steady state manifold. The larger the timescale separation, the larger the gap. Since there can be many separated timescales in a network, we expect correspondingly many gaps. Of note, these gaps separate all the PCs into timescales, with the largest PCs' span representing the infinitely slow timescale.

## Accounting for unobservables: Net reactions

MAK assumes well-mixed, elementary reactions involving the collision of molecules, but single-cell experiments never provide data on all, or even most, of the chemical species participating in

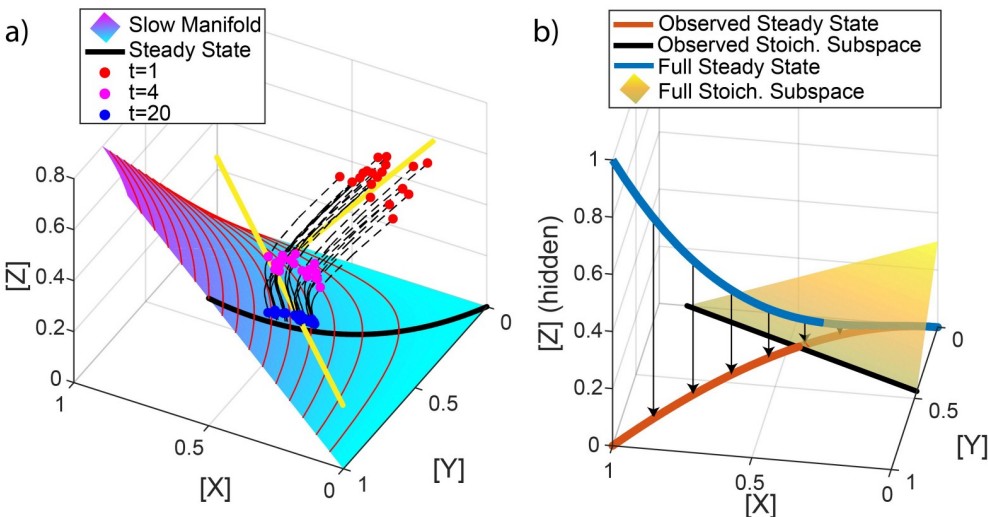

**Fig 2. Timescale separation and hidden variables.** (a) Simulated trajectories are shown for the reaction network with the additional, slow reaction $X \rightleftharpoons Y$ with $k_f = 0.2$, $k_r = 0.1$. Trajectories first converge toward the steady state set of the fast reactions alone, the slow manifold, before slowly converging to the complete steady state (black). (b) An example of a log-linear steady state set (blue, parameterized as $(t, t^2, t^3)$) and its stoichiometric subspace (yellow) are depicted. Supposing we observe $X$ and $Y$, but $Z$ remains hidden, we see the projected steady state set (orange), which is still log-linear. The orthogonal complement we would observe in log-concentration space is the intersection (black) of the original stoichiometric subspace and the observable plane.

elementary reactions for any given biological process. However, assuming that MAK adequately describes the elementary reactions, our conclusions change minimally after accounting for these unobserved species, thanks to log-linearity. More specifically, given a complex-balanced MAK network $G$ that describes the dynamics of $N$ chemical species in $\mathbb{R}^N$, with stoichiometric subspace $S$ and steady state set $\mathcal{E}$, suppose that only a subset $n$ of the species in $N$ is observable. The observed set $\mathcal{E}_{\text{obs}}$ is then the orthogonal projection of $\mathcal{E}$ onto $\mathbb{R}^n \oplus 0 \subset \mathbb{R}^N$, and is still a log-linear set. The orthogonal complement $S_{\text{obs}}$ of $V_{\text{obs}} \equiv \log(\mathcal{E}_{\text{obs}})$ is precisely:

$$S_{\text{obs}} \oplus 0 = S \cap (\mathbb{R}^n \oplus 0). \tag{4}$$

That is, $S_{\text{obs}}$ is the intersection of the stoichiometric subspace and the observable space (see Methods).

As an example in Fig 2b, suppose $N = 3$ chemicals $X$, $Y$, $Z$ obey MAK, but we only observe $n = 2$ of them, $X$ and $Y$. If the steady state set $\mathcal{E}$ is a one-dimensional, log-linear curve, in blue, then $S$ is a two-dimensional plane. Thus, in the observed $\mathbb{R}^2$, we see the projection of $\mathcal{E}$, $\mathcal{E}_{\text{obs}}$, in orange, whose orthogonal complement in log-concentration space, $S_{\text{obs}}$ shown in black, is the intersection between the plane $S$ and the observed plane $\mathbb{R}^2 \oplus 0$.

The fact that the observed orthogonal complement $S_{\text{obs}}$ is a subset of $S$ is important. It implies that any $\vec{v} \in S_{\text{obs}}$ is a linear combination of the reaction vectors that span $S$. Intuitively, a linear combination of elementary reactions is a *net* reaction, just as glucose metabolism is often summarized by

$$\text{Glucose} + 6\,\text{O}_2 \longrightarrow 6\,\text{H}_2\text{O} + 6\,\text{CO}_2,$$

representing a sum of all the elementary reactions that occur during glycolysis and electron transport. As a further example, consider the earlier reaction in (3) and suppose that we only observe $A$ and $B$. The steady state set $\mathcal{E}$ is a line through the origin in $\mathbb{R}^3$, which is still a line

after projecting into the observed $\mathbb{R}^2$. The log-transform of any line through the origin in $\mathbb{R}^2$ becomes a shifted line spanned by $(1, 1)$, whose orthogonal complement is spanned by $(1, -1)$ (see S1 Appendix). Thus, by observing the projected line, and assuming complex-balancing, we can conclude that $A \rightleftharpoons B$ is a net reaction in the full system, which indeed it is: the one direction is given by $A \longrightarrow B$, while the reverse direction is given by $B \longrightarrow C \longrightarrow A$.

In summary, not only is $S_{\text{obs}}$ composed of net reaction vectors, the equality in Eq 4 of $S_{\text{obs}}$ with the intersection implies that $S_{\text{obs}}$ contains *every* net reaction that can be written in terms of the observed chemical species. In this sense, it is *maximal*.

## Networks other than those with complex balance may still have toric geometry: The Effective Stoichiometric Space (ESS)

Whereas complex-balanced networks provide a sufficient condition for the previous results to hold, similar results hold for a larger class of MAK networks, relying on the log-linearity of steady states.

For example, take a simplified kinase($E$)-phosphatase($F$)-substrate($S$) system:

$$E + S \xrightarrow{k_f} E + P$$

$$F + P \xrightarrow{k_r} F + S$$

where the product $P$ has steady-state:

$$0 = \frac{d[P]}{dt} = k_f[E][S] - k_r[F][P] . \tag{5}$$

The complexes $E + S$ and $F + P$ are both reactant complexes, and they appear with opposite sign in the total rate of change of the species $P$. We therefore expect that the orthogonal complement should contain a vector denoting the difference between these complexes. By rearrangement, we see that:

$$\frac{k_r[F][P]}{k_f[E][S]} = 1$$

$$\log(k_r/k_f) + \log([F][P]/[E][S]) = 0$$

$$\log(k_r/k_f) = \log(E) + \log(S) - \log(F) - \log(P) .$$

The orthogonal complement contains $(1, 1, -1, -1)$, which would be seen in data, informing us that $E + S$ and $F + P$ are reactant complexes that balance each other. The result is unchanged if we include the usual Michaelis-Menten enzyme-substrate complex, which is implicit in [29]. Thus, applying our method to data generated by a reaction network that has log-linear, or "toric", steady states, the singular eigenspace still informs us about reaction topology.

Pérez-Millán et al. provide a sufficient condition for a reaction network to have "toric steady states" [29]. This broader class of networks even allows for multistability, which is strictly prohibited for complex-balanced networks. As in the example, the orthogonal complement $V^{\perp}$ of steady states in log coordinates need not coincide with the stoichiometric subspace, although $V^{\perp}$ still relates to network topology.

Furthermore, the steady state set need only be a *subset* of a log-linear set in order to extract the same information, although the set of reactions we recover is no longer maximal. Taking the previous example, add the reactions

$$2X \xrightarrow{k_x} E \qquad Y \xrightarrow{k_y} E \qquad E \xrightarrow{\delta} 0,$$

which imposes an additional, non-log-linear constraint on steady states, to the one in Eq 5:

$$0 = \frac{d[E]}{dt} = k_x[X]^2 + k_y[Y] - \delta[E].$$

Despite this, the previous log-linear constraint in Eq 5 still ensures that at steady state, a sample of trajectories will have zero variance along $(1, 1, -1, -1)$ in the log-coordinates, as in the previous example.

Although some classes of non-complex-balanced have been treated analytically [31, 36–38], we relied on simulation to study non-toric reaction networks in the context of sc-data. As a reference, we first simulated complex-balanced reaction networks with 20 chemical species, including random single and binary reactions. Timescale differences were included by drawing the kinetic constants from two separate distributions (See Methods Table 1). At different timepoints, the distribution of chemical trajectories was subjected to PCA (see Fig 3a). The eigenvalue spectra were found to exhibit gaps that grew larger with time. To confirm that the singular eigenspace spanned the defined stoichiometric subspace, we used Principal Angle Decomposition (PAD) to measure the difference in angles between the two subspaces [39]. We found that the angles converged to zero over time. The slower reactions led to distinctly larger eigenvalues, whose corresponding eigenvectors converged later. Such an example is shown in Fig 3a, where the randomly generated network has a stoichiometric subspace of dimension 11, and the 11 PCs' span converges to the subspace, as evaluated by principal angles. Some reactions were slower, leading to slower convergence along 2 dimensions, visible in the inset. This is accompanied by 2 of the 11 eigenvalues being distinctly larger than the rest, as expected from our previous discussion of how timescale separation manifests as differences in variance.

Having confirmed our conclusions about single-cell data covariance on a complex-balanced simulation, we turned to a non-complex-balanced model. We simulated a Gene Regulatory Network (GRN) with $n$ genes $G_i$, $n$ corresponding protein products $P_i$, and $\sim$70% (chosen randomly) of the possible $n^2$ protein-bound genes $G_i^j$ ($i$'th gene bound by the $j$'th protein) corresponding to proteins that function as transcription activators and repressors (See Methods Table 1). The reactions in the network consisted of irreversible processes that resulted in protein production/degradation, and reversible binding of regulatory proteins to genes:

$$G_i \xrightarrow{k_i} G_i + P_i$$

$$G_i + P_j \underset{r_{ij}}{\overset{f_{ij}}{\rightleftarrows}} G_i^j \xrightarrow{k_i^j} G_i^j + P_i$$

$$P_i \xrightarrow{\delta} 0.$$

**Table 1. Simulation parameter distributions for randomized Complex-Balanced Networks (CB) and Gene Regulatory Networks (GRN).**

| Parameter | Log-Mean | Log-Variance |
|---|---|---|
| Concentration @t = 0 (CB) | 4 | 4 |
| Kinetic Constants (CB) | 2.5, 3 | 0.05 |
| Concentration @t = 0 (GRN) | 5 | 8 |
| Unbound Production Constants (GRN) | 1 | 1 |
| Bound Production Constants (GRN) | 3 | 3 |
| Protein Binding-Unbinding Constants (GRN) | 3 | 1 |
| Protein Degradation Rate (GRN) | 3 | none |

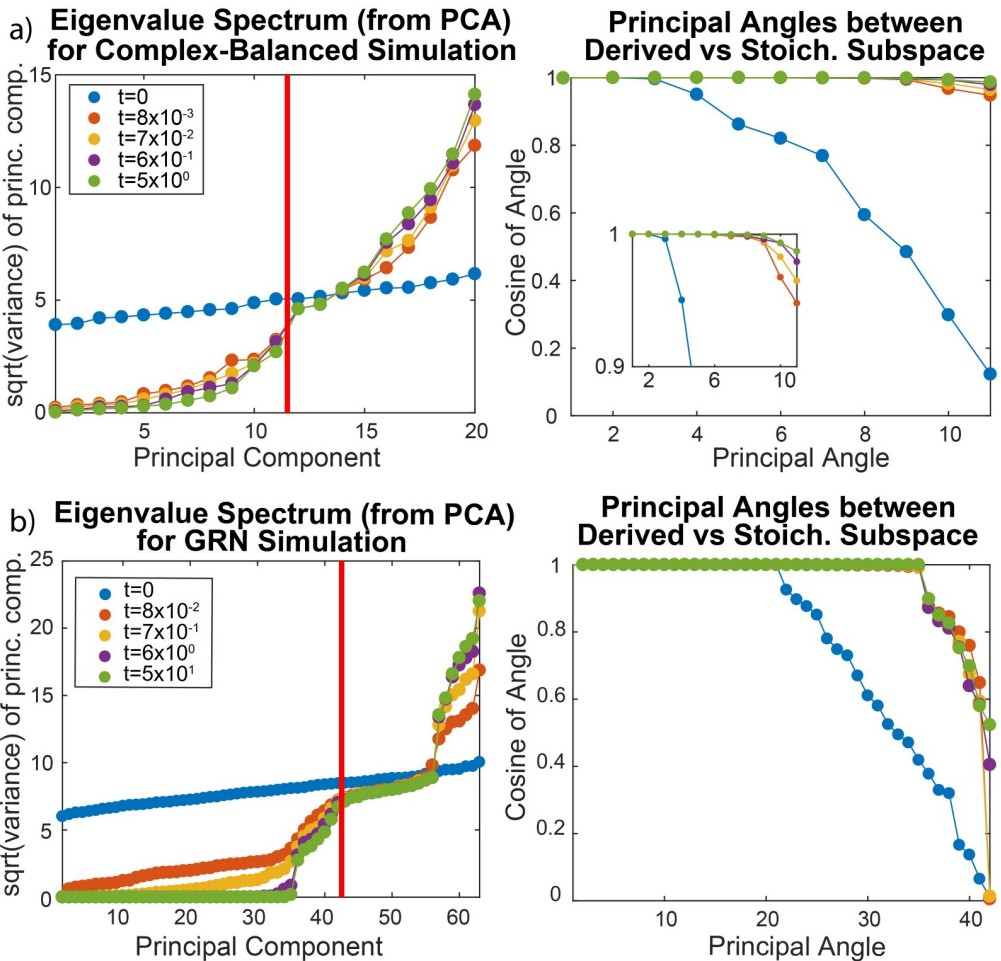

**Fig 3. Reaction network simulations and deriving stoichiometric subspaces.** (a) Example complex-balanced simulation, analyzed by PCA, shows 11 small eigenvalues, as expected from the simulated network's structure, leading to a gap (red line) that grows larger with time. PAD shows that the span of these 11 eigenvectors converges to the true stoichiometric subspace. The 10th and 11th eigenvalues decrease slower than the others, due to slow reactions in the simulation. (b) An example GRN simulation for $n = 7$ is shown. From PCA, a gap in eigenvalues occurs at the expected dimension of the stoichiometric subspace (red line), as well as after the 35th eigenvalue. From PAD, the first 35 eigenvectors span the same subspace as the reversible binding reactions. The remaining 7 eigenvectors before the gap, whose eigenvalues are not as small, span a log-linear space tilted away from the stoichiometric subspace by angles between $\pi/6 \sim \pi/3$. Simulation parameters and networks for both examples are provided in S1 Data.

Transcription/translation was lumped into a single, protein production step for sake of simplicity. The interpretation of this assumption is that mRNA turnover is faster than protein turnover, which is not biologically unreasonable [40]. Analysis demonstrated that networks of this type are indeed non-complex-balanced (see Methods).

When the distribution of trajectories at the end of the simulations was analyzed by PCA, the eigenvalues of the covariance matrix for all simulations exhibited gaps visible in Fig 3b, indicating log-linear constraints. One gap always occurred after $d$ eigenvalues, corresponding to the stoichiometric subspace's dimension as computed symbolically. To evaluate the gap after $d - n$ eigenvalues, we performed PAD on the first $d - n$ eigenvectors and the stoichiometric subspace for the subnetwork of reversible binding reactions, finding that all principal angles were near 0. However, the remaining $n$ eigenvectors converged to a subspace tilted $\pi/6 \sim \pi/3$ away from the stoichiometric subspace.

To understand the convergence of the reversible reactions and the $n$-dimensional tilt, consider a simple example of 2 genes' concentrations, $g_A, g_B$, that of their protein products $p_A, p_B$, and one protein-bound gene, $g_A^B$. In this case the steady state equalities include:

$$\frac{dg_A}{dt} = -\frac{dg_A^B}{dt} = rg_A^B - fg_A p_B = 0$$

from which we retrieve the reaction vector for the reversible binding of protein $B$ to gene $A$. Now, setting the rate of change of protein $B$ to zero, by substitution we have:

$$\frac{dp_B}{dt} = k_B g_B - \delta p_B + rg_A^B - fg_A p_B = k_B g_B - \delta p_B + (0) = 0,$$

giving an orthogonal vector that connects $g_B$ and $p_B$ in a 1 and $-1$ ratio, even though this is not a reaction vector. Finally, we have

$$\frac{dp_A}{dt} = k_A g_A + k_A^B g_A^B - \delta p_A = 0,$$

which does not give a log-linear relation. However, in various limiting cases, it is still possible to recover an asymptotically log-linear relation. For example, consider the common scenario in which protein $B$ is an activator for gene $A$, so that $k_A^B \gg k_A$:

$$\log(k_A^B g_A^B + k_A g_A) = \log(\delta p_A)$$

$$\log(k_A^B g_A^B) + \log(1 + \frac{k_A g_A}{k_A^B g_A^B}) = \log(\delta p_A),$$

and for small $\epsilon = k_A g_A / k_A^B g_A^B$, we recover a log-linear relation by a Taylor expansion of the middle term to zero'th order:

$$\log(k_A^B g_A^B) + \frac{k_A g_A}{k_A^B g_A^B} \sim \log(\delta p_A)$$

$$\log(g_A^B) - \log(p_A) \sim \log(\delta / k_A^B) - \epsilon.$$

This possibly explains the origin of the $n$ eigenvectors that we observe to be tilted relative to the stoichiometric subspace: there are $n$ such $\frac{dp_i}{dt}$ terms in the simulation, each giving an orthogonal vector $\sim (1, -1)$ (the first coordinate being the $i^{th}$ protein species and the second being the most active bound-state of the $i^{th}$ gene), which is tilted $\pi/4$ from the $i^{th}$ protein's reaction vector $(1, 0)$. In our simulation, multiple protein-bound variants existed for any gene, which adds $\epsilon$ error terms that may skew the angles further.

From this one small example, we see that log-linear constraints arise from complex-balanced reactions, from a balance between production and degradation, and from a biological, asymptotic case. We expect log-linear constraints to be mechanistically informative, even without complex-balancing, and thus our framework may be useful with further development in the analysis of general biological systems.

In the remainder of this paper we refer to the orthogonal complement of the minimal, linear set containing the log steady state as the Effective Stoichiometric Space (ESS). The previous examples demonstrate the potential value of the ESS for mechanistic analysis of biological systems, which may often be considered non-complex-balanced. At the very least, if one wants to constrain an asymptotically stable reaction network model using single-cell, multiplex data, a data-derived ESS identifies log-linear relations that must appear in the model's dynamical

equations; this is a fairly precise constraint, since generic polynomial equations are seldom log-linear.

## Single-cell data obtained by FACS has sparse covariance with integer structure

We analyzed a previously published multi-parameter Fluorescence-Activated Cell Sorting (FACS) dataset in which the levels of 11 phospho-proteins in the ERK/Akt signaling pathway were measured in naive primary human CD4+ T-cells [16]. Measurements were made in the presence of 14 different inhibiting or activating perturbations of the pathway. One of the perturbing conditions contained no signal for some phosphorylated species (most likely for technical reasons), so we did not include that condition in our analysis.

FACS data from each condition were fit with a two-component Gaussian Mixture Model (GMM) to distinguish two empirical subpopulations, and the larger component was analyzed further. For each condition, the covariance matrix was eigendecomposed. Each eigenvalue spectrum showed at least one gap, denoted by an orange arrow in Fig 4a; in some cases an additional gap was visible, attributable to timescale separation.

Gaps in eigenvalue spectra were identified by visual inspection, based on the presence of abrupt discontinuities, but the approach is not rigorous. Principled methods do exist to identify which gaps are significant [33], but these methods apply only in the asymptotic limit when the number of dimensions $d \to \infty$, with assumptions on the noise distribution. Thus multiple heuristic methods have been developed, such as looking for spikes in the slope of the spectrum, to choose component numbers in PCA; an overview and comparison of some methods is given in [41, 42]. In the current work we used a heuristic approach to gap identification; our results could potentially be improved with future research into more automated and principled approaches to analyzing eigenvalue spectra.

Each ESS was defined by choosing the gap farthest right. The corresponding eigenvectors were then interpreted by linearly recombining them by row reducing their transpose with complete pivoting [43]. This made it possible to represent the same linear subspace with sparser vectors whose entries are normalized to an arbitrarily selected chemical species. The resulting vectors for a particular condition, are shown in Fig 4c, with a red entry with value 1 denoting the algorithm's chosen normalizing species in each column. Each column can be interpreted as an effective, net reaction, in the broader sense. These data-derived ESS, for each condition, partitioned the 11 markers into groups, implying that steady-state values of chemical species stoichiometrically constrain each other only within these groups (see S1 Fig). The grouping from one condition was identical to that originally generated by Bayesian Causal Network Inference in Sachs et al. (unfortunately, the specific condition is not identified in the manuscript, see [16] Supplement). While the method in [16] pooled the 14 perturbations to infer causal directions, our framework regards each perturbation as a change in the equilibrium constants and topology of the network, without imposing causal structure.

Additionally, the recombined eigenvectors' entries (excluding the 1s and 0s necessarily produced by row reduction) had a distinct distribution (see Fig 4b). First, most entries were near zero (i.e. distributed between -0.2 and 0.2) which suggests a nontrivial sparseness in the span of the selected principal components. The asymmetry of the distribution is also unexpected (see Methods), but is a consequence of our framework: most reaction vectors involve both production and consumption, whose entries necessarily have opposite signs, so after eliminating the positive values of 1 generated by the row reduction algorithm, the remaining nonzero entries of those vectors should always include negative entries. This left 83 entries smaller than

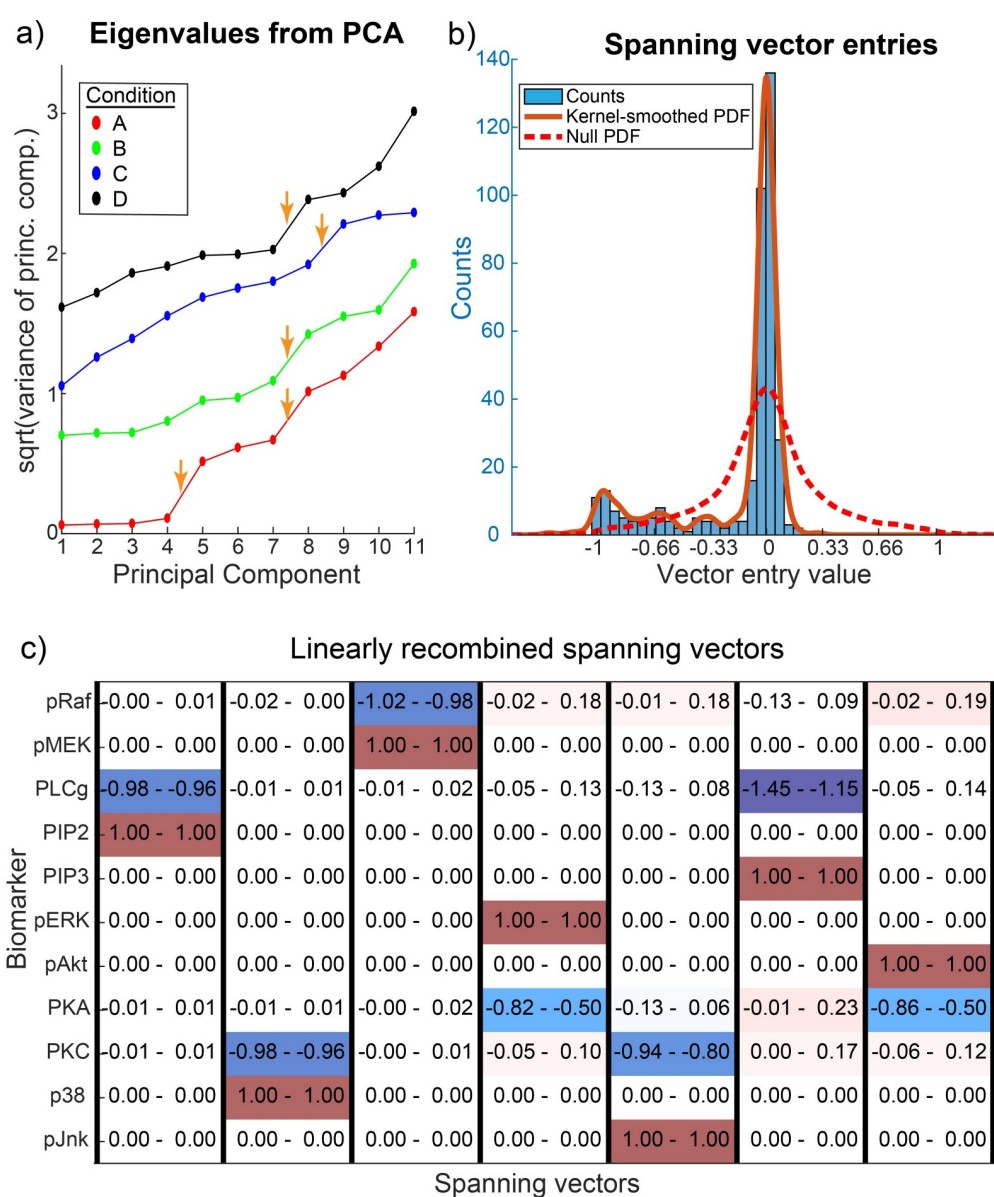

**Fig 4. Peculiar properties of single-cell high-dimensional datasets (FACS).** (a) Eigenvalue spectra from PCA of the larger CD4+ subpopulations are shown for 4 of the 13 conditions (shifted to avoid overlap). Apparent gaps denoted by orange arrows. (b) The small eigenvectors were linearly recombined by row reduction on their transpose, with complete pivoting, for ease of interpretation. The distribution of the linearly recombined entries from all 13 conditions are shown in a histogram (not including the 0s and 1s that are necessarily produced by row reduction), as well as with a Gaussian smoothing kernel of bandwidth 0.04. Peaks seem to appear at -1/3, -2/3, and -1. The null distribution for random, sparse, constraints is also shown for comparison. (c) As an example, the recombined vectors for Condition A are shown, with bootstrapped 95% confidence intervals. Other conditions are similar in appearance.

−0.2 that in our framework are expected to correspond to small-integer-ratios, since reactions typically have small-integer stoichiometries. We notice that the entries bias towards -1/3, -2/3, and -1. These features are significantly different from the null expectation of a random and sparse structure underlying the data, even when accounting for how we heuristically chose the gaps (See S2 Appendix).

## CyCIF data covariance is also sparse and integer-like

We also analyzed a Cyclic Immunofluorescence (CyCIF) dataset that comprises measurement of the levels and modification states of 26 antigens with a focus on phospho-states of proteins involved in apoptosis, Akt/Erk signaling, cell cycle progression, and cytoskeletal structure. The dataset is found in the Library of Integrated Network-based Cellular Signatures (http://lincs. hms.harvard.edu/db/datasets/20267/). Nontransformed MCF10A mammary epithelial cells were exposed to four different kinase inhibitors at six doses each, totaling 24 distinct conditions. Data from each condition were fitted by a GMM with $1 \sim 4$ components, where one component was always substantially larger than the rest; we refer to this component as *dominant*, and focused our analysis on it.

The eigenvalue spectra for each of the conditions also exhibited gaps, as denoted by orange arrows in Fig 5a. Defining each ESS using the gaps around the 10th-14th eigenvalues, row reduction of the selected eigenvectors once again generated the sparse, asymmetric distribution of vector entries observed for FACS data, with a bias to integer-ratios of $-1$, and possibly $-1/2$ (see Fig 5b), although less clear than in the FACS case. For each condition, the row reduced vectors suggest net reactions that sensibly related the proteins involved. For example, in one condition shown in 5c, total amount of S6 protein was linked with that of mTor, and phosphorylated S6 at site S235 was linked to phospho-S6 at site S240, which matches the canonical picture that these proteins influence one another in the mTor-S6 signaling cascade [44]. However, E-Cadherin's contribution to the vector linking S6 with mTor, and the vector linking gamma-H2AX with PCNA, are less expected. The former may reflect the effect of mTor on the Epithelial-mesenchymal transition (EMT) [45], and the latter may reflect the involvement of S phase (as scored by PCNA) and gamma-H2AX in DNA repair. These biological details will require further analysis but the key point is that single-cell microscopy (CyCIF) data resembles FACs data with respect to sparsity, integer-ratio entries, and the appearance of sensible connections between sets of proteins.

## Data from drug-treated cells conserve covariance structure over large dose ranges

For the CyCIF data, we analyzed the dose and drug dependence of the ESS associated with the dominant fluorescent signal in each channel. To compare the subspaces from any two conditions, we first performed PAD between all pairs of conditions, and then summarized the principal angles $\theta_i$ with the metric

$$d = \left( \sum_i \theta_i^2 \right)^{\frac{1}{2}}$$

that appeared in [46]. To interpret this metric, the subspaces being compared must have the same dimension. Thus, we could not use our previously chosen gaps in the eigenvalue spectrum to define ESS for comparison between two conditions; instead, we chose the first 10 eigenvectors in each condition to span a rough ESS for inter-condition comparisons.

Between conditions that did and did not involve exposure to drug (DMSO-only control samples), the ESS changed substantially, as shown in Fig 6a. This is expected, since the addition of a kinase inhibitor alters the set of reactions in the network, potentially altering the ESS. As a toy example, consider the kinase-phosphatase-reaction system described earlier, but with

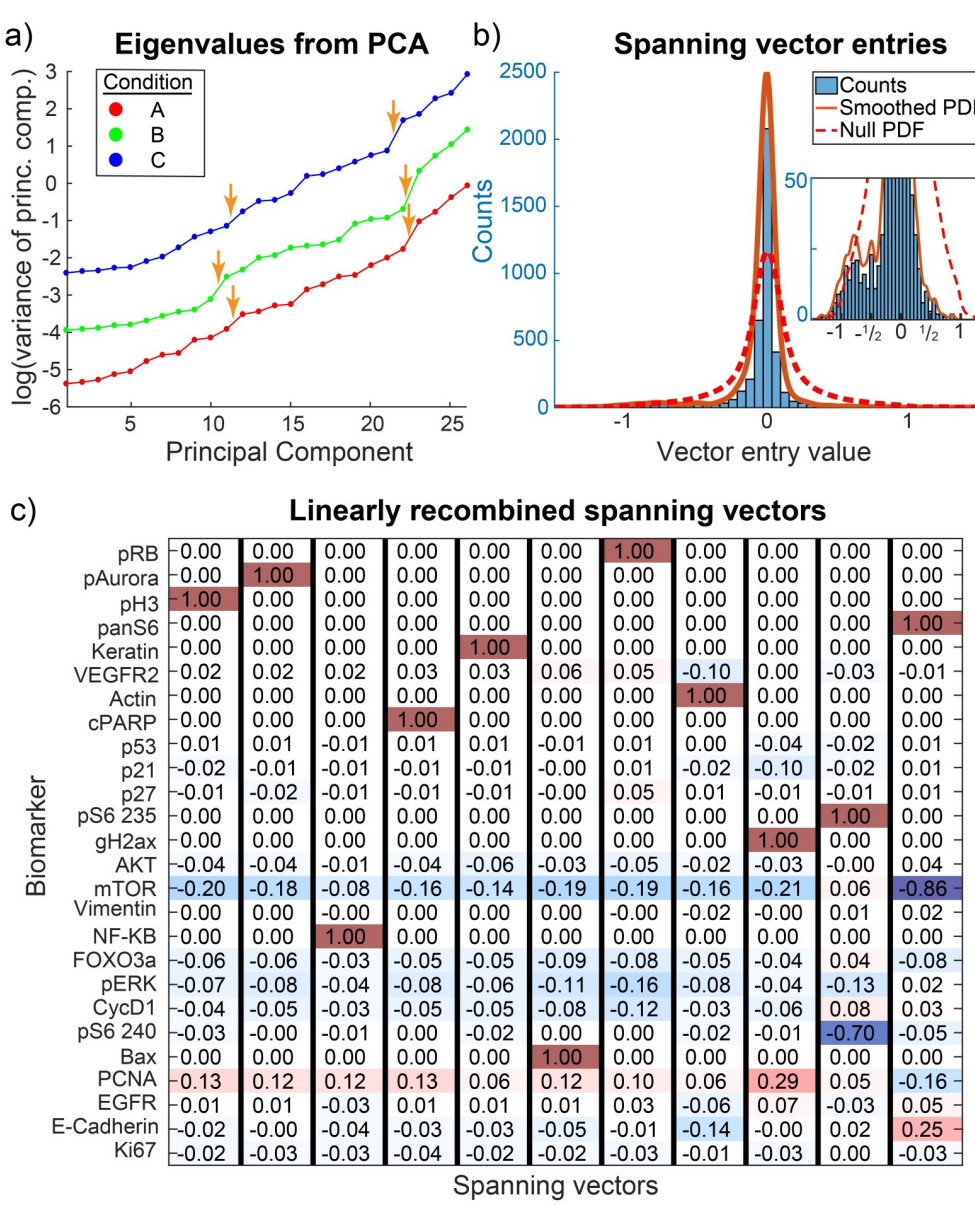

**Fig 5. Peculiar properties of single-cell high-dimensional datasets (CyCIF).** (a) The dominant subpopulations of the MCF10A cells were analyzed by PCA and the eigenvalue spectra are shown for 3 of the 24 conditions (shifted to avoid overlap). Some, but not all, apparent gaps denoted by orange arrows. (b) Singular eigenvectors were linearly recombined by row reduction on their transpose, with complete pivoting. The distribution of entries is displayed, along with a null. (c) Condition A's dominant subpopulation's recombined, singular eigenvectors are shown. Net reactions link the various proteins, such as S6 with mTor, or the two phosphoforms of S6 (235 and 240). Other conditions show similar sparseness.

two distinct enzymes $E_1$ and $E_2$ having the same substrate $S$ and product $P$:

$$E_1 + S \xrightarrow{k_1} E_1 + P \qquad E_2 + S \xrightarrow{k_2} E_2 + P$$

$$F + P \xrightarrow{k_r} F + S.$$

The steady state set is constrained by the non-log-linear relation:

$$\frac{d[S]}{dt} = -k_1[E_1][S] - k_2[E_2][S] + k_r[F][P],$$

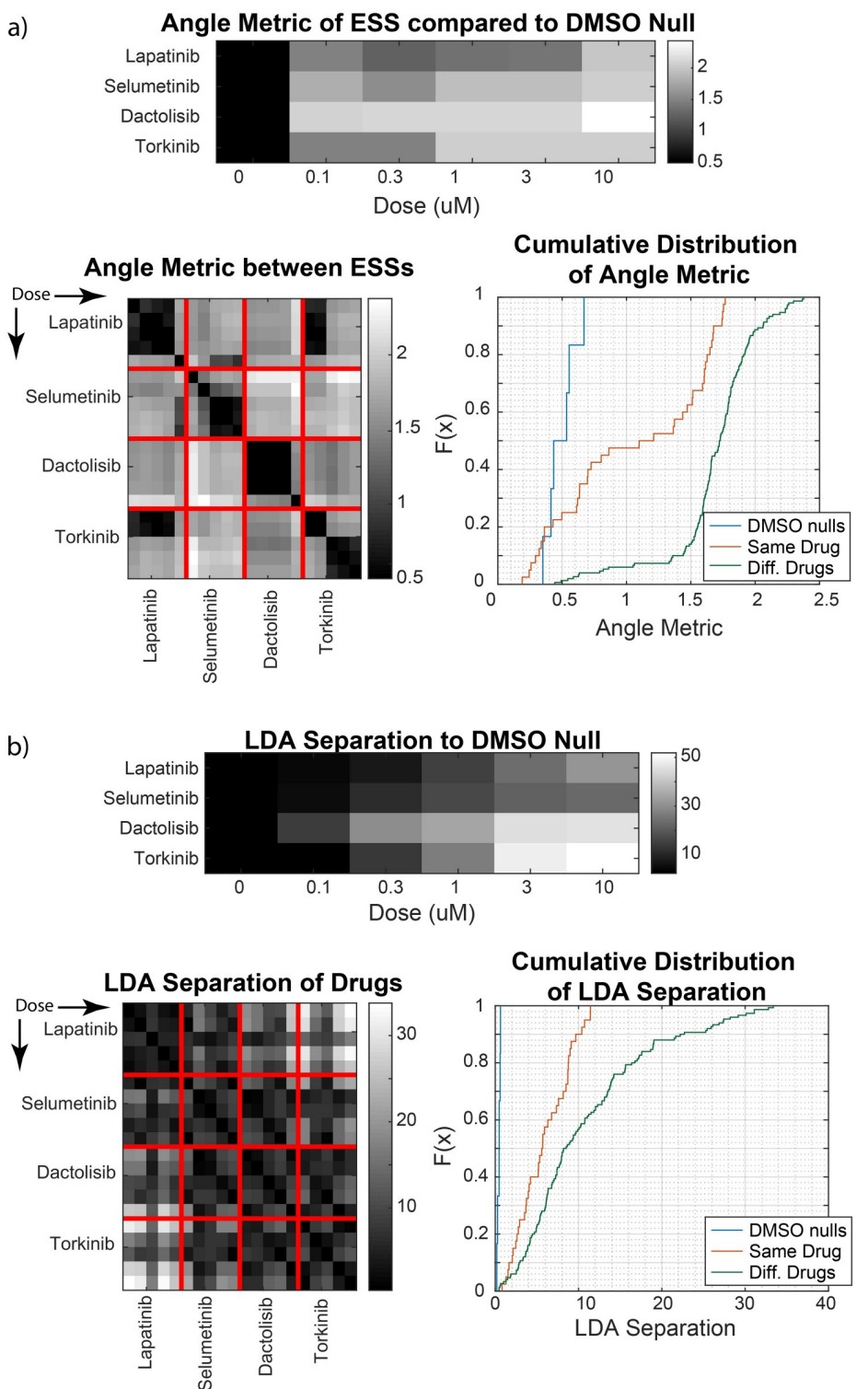

**Fig 6. Comparison of reaction networks between drug conditions.** (a) Analyses of the ESS between conditions, as quantified by an angle-based metric for a common cutoff of a 10-dimensional ESS. Average metric between a drug condition and the four DMSO replicates (top) are plotted, as well as between any pair of non-zero doses of drug (bottom), arranged by increasing dose. The cumulative distribution of the metric is shown for the pairs between DMSO null replicates, the pairs that used the same drug, and the pairs that used different drugs. (b) Analyses of the LDA separation between the high-dimensional marker distributions, with analogous comparisons as above.

but in the limit $k_1[E_1] \gg k_2[E_2]$, i.e. one kinase being much more active than the other, we would observe the approximate log-linear relation $k_1[E_1][S] = k_r[F][P]$ as the ESS. However, by adding a drug $D$ that decreases the active form of $E_1$, such as by inhibition of an upstream activator or $E_1$ itself, we can reach the other limit $k_1[E_1] \ll k_2[E_2]$ at high enough doses. Then, the observed ESS would change to $k_2[E_2][S] = k_r[F][P]$.

Between different doses of the same drug, the data shows the ESS remaining independent of dose almost 50% of the time with a precision comparable to experimental error (based on comparisons between the DMSO-only control samples), as shown in the cumulative distribution of Fig 6a. Between drugs, the ESS differed substantially: $\sim 95\%$ of comparisons showed a larger difference than the null. This is consistent with the following hypotheses: 1) changes in the dose of a drug that inhibits a protein kinase will modify kinetic constants (the rates of substrate phosphorylation) or protein abundances, which is only expected to change the ESS in asymptotic limits, and 2) different drugs interact with different enzymes of the network, resulting in different ESS. The dissimilarities of ESS that do occur between the same drug, do so between dose regimes as opposed to randomly, which is consistent with dose-dependent asymptotics. In the case of Torkinib, this may correspond to its reported polypharmacology [47]. Meanwhile, low-dose Lapatinib and low-dose Torkinib show similar ESS, as do high-dose Lapatinib and high-dose Selumetinib. This implies that in these corresponding dose-regimes, the drugs have similar effects on the 26 observed biochemical species' pathways, which is plausible given that Torkinib's target (mTOR) and Selumetinib's target (MEK) are in two pathways downstream of EGFR, which is one of the targets of Lapatinib. These biological interpretations of the data remain preliminary, but the ESS does appear to pinpoint topological changes in reaction network stoichiometry, independent of parameter changes.

As a more conventional analysis of differences between distributions, we also compared conditions using the Linear Discriminant Analysis (LDA) [48] separation between the dominant Gaussian components of any pair of conditions. The results are shown in Fig 6b, using the same format as the ESS angle-metric comparisons. Almost no pairs have a separation of comparable magnitude to experimental noise, and the magnitude of difference within the same drug is comparable to that between different drugs. Thus, it does not appear that LDA can discern the topologies of reactions perturbed by drugs in the same way as ESS. However, the LDA results do tell us that between drug conditions, the mean marker expression of the dominant populations change substantially. Changes in the mean correspond to changes in the equilibrium constants $K_i$ from Eq 2. Thus, the LDA result showing dose-dependent shifts, combined with the result of dose-independent ESS, allows for a rigorous interpretation: changing drug dose induces parallel shifts of the high-dimensional distribution of cells in marker-space, and changing the drug induces tilts of the distribution.

## Discussion

Single-cell, multiplex imaging and flow cytometry (or mass cytometry [7]) are increasingly used to identify cell states and study regulatory mechanisms. A range of computational methods have been developed to analyze the resulting high-dimensional data but most approaches are statistical. In this paper we explore the possibility of using insights from algebraic-geometry developed for Chemical Reaction Network Theory (CRNT) in the analysis of single-cell data. We find that an effective stoichiometric space (ESS) can be generated from such data to guide reconstruction of biochemical networks. In an initial test of our approach, interpretable network features were obtained from both synthetic and real experimental data. The advantage of using CRNT in this setting is that it provides a principled way to incorporate fundamental knowledge about how biomolecules interact through time and space.

A characteristic of sc-data is that a subset of measured features (typically the levels, localization and modification states of genes and proteins) are observed to co-vary in individuals cells. In the face of random fluctuation, patterns of covariance potentially contain information on interactions between biochemical species. A key question is how this covariance information should be analyzed to obtain insight into the underlying biochemical pathways. We find that eigendecomposition of covariance matrices from sc-data can be interpreted in terms of network stoichiometry and timescales, without model simulation, independent of kinetic parameters, and unhindered by unobserved species; the latter point is critical because most single-cell data is sparse with respect to the number of reactants than can be measured. These features of the ESS approach are a direct consequence of toric (log-linear) manifolds that arise from an assumption of mass-action kinetics applicable—at least approximately—to a broad class of networks, and hold even under the looser requirement that a steady state is a subset of an approximately toric manifold. If the steady state set fulfills this requirement, the ESS also requires that the steady states be asymptotically stable, and that single cells have reached the exponentially stable neighborhood of these steady states at the time of observation. This is looser than requiring that cells be at steady state or quasi-steady state, further expanding the situations in which the ESS framework is informative and applicable.

We tested our approach using synthetic data derived from various simplified reaction systems and also showed that it can be applied to FACS and multiplex imaging datasets. We extract features from the data that are consistent with an interpretation in a reaction network framework: integer-like stoichiometries for interacting species, and independence of network topology on the dose of a single drug used to perturbed the network. Other kinds of sc-data, such as mass cytometry or sc-RNAseq, can potentially be analyzed using the same approach. Because this paper focuses on the theoretical aspects of toric geometries as applied to sc-data, we have not yet tested any of the biological conclusions derived from the analysis of experimental data. However, the interactions we infer are consistent with current understanding of well-studied human signal transduction networks and with previous publications [16]. More extensive single-cell experiments will be required to fully test the potential for ESS analysis to generate new biological insight.

Simulation of synthetic complex-balanced networks and GRNs suggests ways to tailor reaction network ODEs to better match sc-data. Assuming that the primary goal in fitting a network to data is to match the mean $\mu$ and covariance $\Sigma$ of key analytes, our results show that it is possible to predict a partitioning of the eigenspace of $\Sigma$ without actually simulating the ODE network, under the assumption of toric geometry. To accomplish this fitting, it is necessary to account for initial conditions to predict $\mu$ and the exact eigendecomposition of $\Sigma$, but this may still be possible in the absence of simulation. Such an approach would not only take advantage of information unique to single cells, but may also make it possible to parameterize models too complex for conventional fitting (this is important because fitting conventionally involves many rounds of simulation and is computation-intensive). Because it has explicit connections to CRNT, such a method could be used in conjunction with other recently developed applications of CRNT for data-constrained, ODE model selection [49–53]. This provides a principled way to choose among models with different components and topologies, a common goal of systems biology modeling projects.

One limitation of the network analysis approach described here is that identifying gaps in the eigenvalue spectrum is a heuristic procedure. Unfortunately, this is true of most other applications in which it is necessary to identify cutoffs in eigenvalue spectra. The relatively low dimensionality of FACS and CyCIF datasets further limits the applicability of those principled approaches that are available, including methods in random matrix theory. However, for larger datasets it will potentially be possible to apply principled methods for identifying gaps that are statistically significant.

Cell regulatory networks are characterized by multistability and limit cycles. The relationship between our analysis and such network structures remains unclear and will require further theoretical work. Multistable, toric steady state sets exist, with many biologically relevant ones studied recently in the class of toric MESSI systems [31], but there are many circumstances in which multistable states are not toric. Perhaps such non-toric sets can be approximated as toric in various limiting regimes, but even then, certain parts of the steady state set must necessarily be unstable to support multistability. For limit cycles, the expected geometry is not necessarily algebraic, although one might hope that the limit cycle is contained in an almost-toric manifold, so that our approach is still informative. Exploring these and other issues requires further development of the connection between the sc-data and CRNT.

The promise of the ESS approach is that it provides a potentially powerful but as-yet unexplored, geometric framework for linking features in sc-data to reaction networks. This is a parallel to recent geometrical analysis of CRNT, in which toric varieties have played a key role. Toric varieties have aided in characterizing the central CRNT concept of complex-balancing [25] and they have also enabled systematic determination of kinetic parameters that give rise to multistability for large classes of networks [54], including biologically relevant networks such as the MAPK pathway [30]. In the context of sc-data, we leverage toric geometry to study the reactions underlying cellular phenotypes without having to perform simulations, which can be difficult with sparse and complicated data. Despite the fact that some network steady states do not necessarily conform to toric geometries, the CRNT framework as accessed through ESS is a closer approximation to the reality of biological networks than the statistical and dimensionality reduction approaches (clustering, tSNE etc.) that currently dominate data analysis. With further development of the ESS approach, it should be possible to use CRNT to formulate mechanistic hypotheses from data in the absence of simulation and then subject the hypotheses to empirical tests.

## Methods

### Sc-covariance matrix from complex-balanced reaction networks

For any complex-balanced reaction network $G$, the steady state set $\mathcal{E}$ is log-linear, so as $t \to \infty$, each $\log(\vec{c}_i(t))$ approaches a linear subset $V = \log(\mathcal{E})$. Therefore, in the limit, the sample covariance matrix is singular, and its singular eigenspace $\mathcal{S} \supset V^\perp$, the orthogonal complement of $V$. Complex-balancing implies that $V^\perp = S$, the stoichiometric subspace of $G$ [25], so $S \subset \mathcal{S}$.

The equality $S = \mathcal{S}$ arises when the distribution of $\vec{v} \cdot \vec{c}_i(t)$, where $i$ indexes over all cells, at $t = 0$ has non-zero variance $\sigma_v(t) \equiv \mathrm{Var}(\vec{v} \cdot \vec{c}_i(t))$ for all $\vec{v} \in V$. Splitting the chemical concentration space into vector-additive cosets of the stoichiometric subspace $S$, all trajectories of a complex-balanced system remain within the coset, by forward-invariance of these cosets. By orthogonality of $V$ to $S$, $\sigma_v(t)$ is time-independent. If $\sigma_v$ is non-zero for all $v$, then the variance in log-concentration space of $\vec{v}$, given by $\vec{v} \cdot \log(\vec{c}_i(t))$ cannot be zero. Therefore, $\vec{v} \notin \mathcal{S}$, so $\mathcal{S} \subset V^\perp$. Together with the previous inclusion, $\mathcal{S} = V^\perp = S$, as $t \to \infty$.

There is a unique steady state point $c_e$ in each coset of a complex-balanced reaction network [24], which has an exponentially stable neighborhood. The Global Attractor Conjecture, for which a proof was announced [55] but has not yet been peer reviewed, suggests that all complex-balanced reaction networks are globally asymptotically stable (relative to a stoichiometric coset), so for sufficiently long $T$, any finite collection of trajectories uniformly enter their exponentially stable neighborhoods. After this time $T$, any two trajectories $c_1(t)$ and $c_2(t)$ on the same coset obey

$$\|c_1(t) - c_e\| \leq \alpha \|c_1(T) - c_e\| e^{-\beta(t-T)}, \qquad \|c_2(t) - c_e\| \leq \alpha \|c_2(T) - c_e\| e^{-\beta(t-T)}$$

and so by the triangle inequality,

$$\|c_1(t) - c_2(t)\| \le \|c_1(t) - c_e\| + \|c_2(t) - c_e\| \le \alpha(\|c_1(T) - c_e\| + \|c_2(T) - c_e\|)e^{-\beta(t-T)},$$

which gives a monotonically decreasing, upper bound on the distance between any pair of trajectories in a coset. As distance between these trajectories decreases, their variance decreases. As long as the steady state set has no boundary states (nontrivial equilibria with zero concentration for some chemical species), the logarithm of concentrations will also have a uniform upper-bound which decreases with time, after some time $T'$.

While the variance along the orthogonal complement of $V$ decreases to 0, the variance along $V$ remains non-zero, so the eigenvalues of the covariance matrix separate into those that are non-zero (large) and those approaching zero (small). Assuming no further network structure, at sufficiently large times, we consider the distribution of trajectories as a spiked population model in which the small eigenvalues follow a Marchenko-Pastur density with rescaled support [33], while the large eigenvalues lie outside its support. This forms a gap.

Furthermore, if the reaction network has a subnetwork with separably faster rates of convergence than the entire network, additional gaps may occur. In the case of detailed-balance reaction networks, this follows from a singular perturbation approach: Giovangigli et al. showed that for separably fast and slow reactions in such networks, the critical steady-state manifold is equivalent to the steady-state manifold of a network containing only the fast reactions [35]. Therefore, trajectories first converge to $V_{\text{fast}} \supset V$, leading to a gap between the fast, small eigenvalues and the slow, larger eigenvalues.

## Log-linearity despite unobservables

Given a $D$-dimensional log-linear set in $\mathbb{R}^N$, parameterize it by a $D$-dimensional vector of parameters $\vec{p}$ and $D$ corresponding column vectors $\{\vec{v}_i\} \subset \mathbb{R}^N$ that span the set s.t.:

$$\log(\vec{c}) = [\vec{v}_1 \dots \vec{v}_D]\vec{p} + \log(\vec{c}^*)$$

for some point $\vec{c}^*$ in the set that determines the translation of the affine set. For $A \equiv [\vec{v}_1 \dots \vec{v}_D]$, the orthogonal complement $S_N$ is the null space of $A^T$.

Split the matrix $A$ horizontally into two matrices, $A_{\text{obs}}$, and $A_{\text{unobs}}$, corresponding to $n$ observed chemical species and $N - n$ unobserved species. The vector whose coordinates are the observed species, $\vec{c}_{\text{obs}}$, are parameterized by

$$\log(\vec{c}_{\text{obs}}) = [A_{\text{obs}}]\vec{p} + \log(\vec{c}^*_{\text{obs}})$$

$$A = \begin{bmatrix} A_{\text{obs}} \\ A_{\text{unobs}} \end{bmatrix}$$

where the chemical species are rearranged for convenience, without loss of generality. Therefore, provided $n > D$, the data still lies in a nontrivial, log-linear set.

Now we show that the orthogonal complement of $A_{\text{obs}}$, $S_n \equiv \text{Null}(A_{\text{obs}}^T)$, is a meaningful subspace:

$$S_n \oplus 0_{N-n} = S_N \cap (\mathbb{R}^n \oplus 0_{N-n})$$

where $0_{N-n}$ is the zero-vector in the $(N - n)$-dimensional unobserved space.

*Proof*: For the forward inclusion, observe that $S_n \subset \mathbb{R}^n$, and simultaneously that $S_n \oplus 0_{N-n} \subset S_N$, because for all $\vec{v} \in S_n$,

$$[A^T](\vec{v} \oplus 0_{N-n}) = [A_{\text{obs}}^T \mid A_{\text{unobs}}^T](\vec{v} \oplus 0_{N-n}) = [0 + 0] = 0,$$

since by definition of $\vec{v}$, $[A_{\text{obs}}^T]\vec{v} = 0$.

For the reverse inclusion, a vector in $(\mathbb{R}^n \oplus 0_{N-n})$ takes the form $\vec{w} \oplus 0_{N-n}$, and being in $S_N$ indicates that

$$0 = [A^T](\vec{w} \oplus 0_{N-n}) = [A_{\text{obs}}^T \mid A_{\text{unobs}}^T](\vec{w} \oplus 0_{N-n}) \Rightarrow [A_{\text{obs}}^T]\vec{w} = 0$$

and therefore $\vec{w} \oplus 0_{N-n} \in S_n \oplus 0_{N-n}$.

## Simulation of random networks

ODE simulations of each reaction network were performed with the *ode*15*s* function in MATLAB. For each network, 300 initial conditions were chosen from a log-normal distribution with equal log-variance for all chemicals. Specific sampling distribution parameters are in Table 1.

**Complex-balanced networks.** All complex-balanced networks were chosen to have $n = 20$ chemical species. Complex-balancing is defined via the digraph with nodes representing *complexes*, such as $A + B$, and edges representing reactions between complexes. A sufficient condition for complex-balancing is for a network to have *deficiency* equal to 0, and be *weakly reversible*. Deficiency $\delta$ is defined as

$$\delta = n - l - s$$

where $n$ is the number of complexes, $l$ is the number of weakly connected components, termed *linkage classes*, and $s$ is the dimension of the stoichiometric subspace. Weak reversibility amounts to all nodes belonging to a strongly connected component.

Each random network was generated with all the nodes representing complexes containing either a single species, or any pair of species. Then, $\sim 0.03\%$ of the possible edges were stochastically chosen. The graph was then symmetrized by adding all the reverse edges, to ensure reversibility. Rate constants were randomly assigned from a log-normal distribution (see Table 1). Many such networks were generated, and only the ones with deficiency zero were simulated.

**Gene regulatory networks.** For the GRNs, for $n$ genes, $\sim 70\%$ of the possible protein-bound genes $G_i^j$ were chosen stochastically. Kinetic constants for each type of reaction were chosen from log-normal distributions whose log-mean and log-variance are shown in Table 1.

The GRN simulations were not complex-balanced, both because the particular arrangement of irreversible reactions violate weak reversibility, and because the deficiency of the networks were large, indicating a measure zero probability of being complex-balanced.

## Linear fluorescence assumption

The framework calls for analyzing chemical concentrations $\vec{c}$. Both FACS and CyCIF data contain fluorescence intensity signals instead of the actual concentrations, but our method still applies if the $i^{th}$ chemical species' signal $I_i = k_i \cdot c_i$ for some constant $k_i$ for the cells in one sub-population. Assuming an excess of antibodies for both the experimental setup of the FACS and CyCIF data, this is simply the requirement that detection is in the linear regime.

The method still works because the $k_i$'s would only result in a shift of the affine subspace $V$. For any log-linear constraint on the $\vec{c}$

$$\log(K) = \vec{r} \cdot \log(\vec{c}),$$

the observed constraint in terms of $\vec{I}$ is

$$\log(K) + \vec{r} \cdot \log(\vec{k}) = \vec{r} \cdot \log(\vec{I}).$$

## Gaussian mixture modeling

Both FACS and CyCIF data were fit with Gaussian mixture models (GMM) to match visible clusters. Cells from any single condition were fit with $k$ components, with $k$ chosen based on abrupt decreases in the incremental likelihood gain for additional components, while also preventing the splitting of visible clusters in the data.

GMMs were fit using the *fitgmdist* function in MATLAB 2016b, allowing the Expectation Maximization algorithm to run to convergence for at least $20k$ different initializations chosen by the $k$-means++ algorithm.

## Null distribution of row reduced vector entries

Assuming sparse, random, linear constraints on a distribution, the covariance matrix would have singular eigenvectors whose span can be given by sparse vectors with random orientations. For either the FACS or CyCIF data, the null was given by row reducing $s$ vectors whose entries were chosen uniformly between −0.5 and 0.5, and subsequently made sparse at random entries. The dimension of the constraints, $s$, was chosen to be similar to that selected for each datasets' stoichiometric subspace, and sparsity was set equal to the percentage of the zero-centered peak of the data's entry-distribution in a window between −0.2 and 0.2. Gaussian noise was added to the final row-reduced vector entries, with variance matching that of the zero-centered window. Both null distributions were generated by Gaussian kernel smoothing of 1000 such sets of $s$ row reduced vectors' entries.

## Confidence intervals for row-reduced vector entries

The 95% confidence intervals for the row-reduced vector entries in Fig 4 were calculated by bootstrapping for 1000 replicates. Each time, the original data was resampled with replacement, before fitting a GMM with the same number of components as for the original data. Then, the stoichiometric subspace was chosen with the same dimension. Finally, row reduction was performed with the same column and row rearrangements as was done in the original data, instead of using the complete pivoting algorithm, to keep entries consistent between replicates.

## Supporting information

**S1 Appendix. Steady state analysis of network (3) with unobservables.** (PDF)

**S2 Appendix. Probability of manual gap choice producing observed vector entry distribution.** (PDF)

**S1 Fig. Interaction network from FACS.** Given recombined, singular vectors for the condition of activation with anti-CD3, anti-CD8 and inhibition of Protein Kinase C with G06976,

we drew an edge between biomarkers if any vector entries had magnitude larger than 0.2. (TIF)

**S1 Data. Simulation parameters and reaction networks for Fig 3.**
(ZIP)

## Author Contributions

**Conceptualization:** Shu Wang, Eduardo D. Sontag, Peter K. Sorger.

**Data curation:** Jia-Ren Lin.

**Formal analysis:** Shu Wang, Jia-Ren Lin, Eduardo D. Sontag.

**Funding acquisition:** Eduardo D. Sontag, Peter K. Sorger.

**Investigation:** Shu Wang, Jia-Ren Lin, Eduardo D. Sontag.

**Methodology:** Shu Wang.

**Supervision:** Jia-Ren Lin, Eduardo D. Sontag, Peter K. Sorger.

**Visualization:** Shu Wang.

**Writing – original draft:** Shu Wang.

**Writing – review & editing:** Shu Wang, Eduardo D. Sontag, Peter K. Sorger.

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
