## [Decision Letter · Decision Letter 0]

17 Sep 2019

Dear Dr Sorger,

Thank you very much for submitting your manuscript, 'Inferring reaction network structure from single-cell, multiplex data, using toric systems theory', to PLOS Computational Biology. As with all papers submitted to the journal, yours was fully evaluated by the PLOS Computational Biology editorial team, and in this case, by independent peer reviewers. The reviewers appreciated the attention to an important topic but identified some aspects of the manuscript that should be improved.

We would therefore like to ask you to modify the manuscript according to the review recommendations before we can consider your manuscript for acceptance. Your revisions should address the specific points made by each reviewer and we encourage you to respond to particular issues Please note while forming your response, if your article is accepted, you may have the opportunity to make the peer review history publicly available. The record will include editor decision letters (with reviews) and your responses to reviewer comments. If eligible, we will contact you to opt in or out.raised.

- Supporting Information uploaded as separate files, titled 'Dataset', 'Figure', 'Table', 'Text', 'Protocol', 'Audio', or 'Video'.

We hope to receive your revised manuscript within the next 30 days. If you anticipate any delay in its return, we ask that you let us know the expected resubmission date by email at ploscompbiol@plos.org.

Sincerely,

Pedro Mendes, PhD

Associate Editor

PLOS Computational Biology

Mona Singh

Methods Editor

PLOS Computational Biology

[LINK]

Reviewer's Responses to Questions

**Comments to the Authors:**

Reviewer #1: Review: Inferring reaction network structure from single-cell, multiplex data, using toric systems theory, PCOMPBIOL-D-19-01320.

I find this article a nice contribution to the literature, it proposes a novel combination of network structure and statistical study of reaction networks. Partial linearities in log coordinates of steady states of the system are exploited. I suggest acceptance after the authors incorporate my comments below in a revised version of their submission.

1) The stoichiometric subspace S is defined as the linear span of the reaction vectors on line 55, which is correct. However, the reaction vector v defined on line 47 is not the true reaction vector whenever some of the reactant R_i coincides with a product P_j, This does not necessarily happen, e.g. this is not the case in the example on line 198. So this is misleading and it becomes an obscure concept for the reader, while it is very easy to compute S if the whole reaction network is known. Please, define S clearly and explain that its computation is straightforward if the network is known.

2) In display (3), it would be better to add the arrow from C to A curved back to the first node A on the left instead of adding a new node A. This weak reversibility is what allows for a complex balanced steady state and this is an equivalent condition for a linear network as this one.

3) On line 214 it seems that you are missing some references, e,g: SIAM J. Appl. Dyn. Syst., 17(2), 1650–1682. The Structure of MESSI Biological Systems. Mercedes Pérez Millán and Alicia Dickenstein. https://doi.org/10.1137/17M1113722

Please, take a look at the notion of toric MESSI systems and how the networks are amenable to analytical treatment without simulations (as in several other recent articles). Another important missing reference is:J Theor Biol. 2009 Dec 21;261(4):626-36. The rational parameterization theorem for multisite post-translational modification systems. Thomson M1, Gunawardena J. doi: 10.1016/j.jtbi.2009.09.003.

4) Please add clearly in all your figures (and their corresponding explanations) which are the values of the reaction rate constants that are being taken into account.

5) On line 232, explain which is the assumption needed in order that the consideration of a single protein production step is sensible.

6) Please, make the definition of EES more visible, add a pointer to it from the very beginning of the paper (and consider giving it before in the text).

7) May be it is useful to take a look at section 5 of the MESSI reference above: when the positive steady states can be defined completely by binomial equations (linear after taking logarithms), the comparison of the matrix built from the exponents of these binomials and a matrix given the linear conservation relations defining S, can be used to detect multistationarity. In this case, the analysis cannot be done for all choices of parameters at the same time, but in regions of mono or multistationarity separately (plus finer stability considerations).

8) Your example on page 12, line 344 is easily seen to be s-toric MESSi. You are missing the fact that d[D]/dt = - k_d [S][D] – k_{-d} [SD], thus adding this to d[S]/dt and equating to 0, one immediately gets two binomials: this new one and the previous one involving [E],[S],[F],[P] describing $E$! (without any limit!). By the way, it is unfortunate that the steady state set is denoted by $E$ while E denotes a chemical species. I’d suggest to change the letter for the steady state set.

Reviewer #2: The paper discusses an important question: given measurements of a set of chemical species, which reaction network can reproduce the measurements sufficiently well. The authors suggest a novel approach that requires single cell data and that the data indicates that the measurements approach a steady state. The idea is to analyze the spectrum of the covariance matrix of the measurements and look for a "gap", indicating a subset of eigenvalues that can be considered zero. The eigenvectors corresponding to the nonzero eigenvalues are then interpreted as "effective reactions" that span an effective stoichiometric space. This space is interpreted as the image of a matrix that parametrizes the positive real part of the steady state variety.

I have only one major concern with the paper, it is not precise enough when it comes to the claim that a reaction network is inferred form the data. Only if one assumes that the system is complex balanced, is the effective stoichiometric space related to the reaction network: steady states of complex balanced systems are parameterized by the orthogonal complement of the stoichiometric space. Hence only in this case one may actually claim to have inferred a reaction network. If the system is not complex balanced, then the relation between the reaction network and the matrix that parametrizes the positive real part of the variety is more involved. I think the authors should describe this more clearly: if one chooses to interpret what is inferred as a reaction network, then the corresponding system is complex balanced (with the all the constraints that imposes on the rate constants). If one does not want to impose complex balance, then one infers a monomial parametrization of the positive real part of the variety (which is remarkable enough).

A suggestion: the approach can be applied to any data set coming from single cell measurement. If one finds a gap in the covariance matrix, then this indicates that the data can be described by a model that has a remarkable property: the positive real part of the variety can be parameterized by monomials. For generic polynomials this is rarely the case, so maybe one could advertise this.

Some minor points:

- the caption of figure one: I think it should be steady state set, not steady state

- line 67: I think the name is Kirchhoff

-line 404: I do not think that log-linear manifolds arise from mass action kinetics. They require a special network structure, not every mass action network gives rise to such a manifold.

-line 406-408. This sentence does not make any sense to me. You say you do not need QSS but that the steady state is asymptotically stable. And then you say that enough time should have passed to approach the QSS What is it that you want to say?

-line 482 what does it mean when you say all trajectories within a coset are forward invariant? It is my understanding that the coset is forward invariant.

-line 488 you could state that the upper bound is for all trajectories within a coset, just to be precise.

-reference 18. As far as I know, the authors describing mass action kinetics for the first time are called Waage and Guldberg.

**Have all data underlying the figures and results presented in the manuscript been provided?**

Reviewer #1: No: The rate constants considered should be clearly specified.

Reviewer #2: Yes

PLOS authors have the option to publish the peer review history of their article (what does this mean?). If published, this will include your full peer review and any attached files.

Reviewer #1: No

Reviewer #2: No

---

## [Editor Report · Decision Letter 1]

8 Nov 2019

Dear Dr Sorger,

We are pleased to inform you that your manuscript 'Inferring reaction network structure from single-cell, multiplex data, using toric systems theory' has been provisionally accepted for publication in PLOS Computational Biology.

In the meantime, please log into Editorial Manager at https://www.editorialmanager.com/pcompbiol/, click the "Update My Information" link at the top of the page, and update your user information to ensure an efficient production and billing process.

One of the goals of PLOS is to make science accessible to educators and the public. PLOS staff issue occasional press releases and make early versions of PLOS Computational Biology articles available to science writers and journalists. PLOS staff also collaborate with Communication and Public Information Offices and would be happy to work with the relevant people at your institution or funding agency. If your institution or funding agency is interested in promoting your findings, please ask them to coordinate their releases with PLOS (contact ploscompbiol@plos.org).

Thank you again for supporting Open Access publishing. We look forward to publishing your paper in PLOS Computational Biology.

Sincerely,

Pedro Mendes, PhD

Associate Editor

PLOS Computational Biology

Mona Singh

Methods Editor

PLOS Computational Biology

---

## [Editor Report · Acceptance letter]

26 Nov 2019

PCOMPBIOL-D-19-01320R1 

Inferring reaction network structure from single-cell, multiplex data, using toric systems theory

Dear Dr Sorger,

I am pleased to inform you that your manuscript has been formally accepted for publication in PLOS Computational Biology. Your manuscript is now with our production department and you will be notified of the publication date in due course.

With kind regards,

Sarah Hammond
